# AI-Enabled Mosquito Surveillance and Population Mapping Using Dragonfly Robot

**DOI:** 10.3390/s22134921

**Published:** 2022-06-29

**Authors:** Archana Semwal, Lee Ming Jun Melvin, Rajesh Elara Mohan, Balakrishnan Ramalingam, Thejus Pathmakumar

**Affiliations:** Engineering Product Development Pillar, Singapore University of Technology and Design (SUTD), Singapore 487372, Singapore; archana_semwal@sutd.edu.sg (A.S.); melvin_lee@sutd.edu.sg (L.M.J.M.); rajeshelara@sutd.edu.sg (R.E.M.); pathmakumar_thejus@mymail.sutd.edu.sg (T.P.)

**Keywords:** robot, mosquito surveillance, deep learning, computer vision, mapping

## Abstract

Mosquito-borne diseases can pose serious risks to human health. Therefore, mosquito surveillance and control programs are essential for the wellbeing of the community. Further, human-assisted mosquito surveillance and population mapping methods are time-consuming, labor-intensive, and require skilled manpower. This work presents an AI-enabled mosquito surveillance and population mapping framework using our in-house-developed robot, named ‘Dragonfly’, which uses the You Only Look Once (YOLO) V4 Deep Neural Network (DNN) algorithm and a two-dimensional (2D) environment map generated by the robot. The Dragonfly robot was designed with a differential drive mechanism and a mosquito trapping module to attract mosquitoes in the environment. The YOLO V4 was trained with three mosquito classes, namely Aedes aegypti, Aedes albopictus, and Culex, to detect and classify the mosquito breeds from the mosquito glue trap. The efficiency of the mosquito surveillance framework was determined in terms of mosquito classification accuracy and detection confidence level on offline and real-time field tests in a garden, drain perimeter area, and covered car parking area. The experimental results show that the trained YOLO V4 DNN model detects and classifies the mosquito classes with an 88% confidence level on offline mosquito test image datasets and scores an average of an 82% confidence level on the real-time field trial. Further, to generate the mosquito population map, the detection results are fused in the robot’s 2D map, which will help to understand mosquito population dynamics and species distribution.

## 1. Introduction

Mosquito-borne diseases remain a significant cause of morbidity and mortality across tropical regions. Thus, mosquitoes are considered a significant problem for human health. The mosquito transmits infectious pathogens to humans through bites, serving as vectors of diverse life-threatening diseases such as dengue, chikungunya virus, dirofilariasis, malaria, and Zika. In Singapore, the National Environmental Agency (NEA) has stated open perimeter drains, covered perimeter drains, covered car parks, and roadside faulty drainage sites as the most common breeding habitats for mosquitoes in public areas [1]. Hence, routine mosquito surveillance is essential for effective control of the mosquito population. According to the Integrated Mosquito Management (IMM) program, mosquito surveillance includes inspecting breeding sites, identifying mosquito types, and measuring the critical environment through mosquito population mapping. However, conventional surveillance methods such as manual inspection are used to detect and classify mosquitoes, which are time-consuming, difficult to monitor, and labor-intensive. Thus, automation of mosquito surveillance and breed classification needs to be a high priority.

Various human-assisted methods have been used for mosquito surveillance in the past decade. For example, thermal fogging trucks, pesticides, and electrical traps have been regularly and effectively used for mosquito surveillance. Further, traps can be divided into active traps (visual, olfactory, or thermal cues) and passive traps (suction fans). In the literature, various mosquito trapping devices, such as the Biogents sentinel trap [2], heavy-duty Encephalitis Vector Survey trap (EVS trap) [3], Centres for Disease Control miniature light trap (CDC trap) [4], Mosquito Magnet Patriot Mosquito trap (MM trap) [5], MosquiTrap [6], and Gravitrap [7], have been reported. These devices use various techniques, such as a fan, contrast light, scent dispenser, carbon dioxide, and magnets, to target mosquitoes and breeding sites. These methods are labor-intensive and consume a lot of time. However, the efficiency of these tracking devices is greatly influenced by the device’s location, time of deployment, number of devices, and the extent and duration of the transmission. The contribution and efficiency of such devices are quite uncertain due to the complex interplay of multiple factors. Lastly, mosquito surveillance and mosquito population mapping require a highly skilled workforce.

Recently, machine learning (ML) approaches have been widely used for mosquito detection and classification tasks. Here, Artificial Neural Network (ANN), Decision Trees (DT), Support Vector Machines (SVM), and Convolutional Neural Networks (CNNs) are the most commonly used methods. Among these techniques, CNN-based frameworks are the more popular mosquito detection and classification tool. In [8], Kittichai et al. presented a deep learning-based algorithm to simultaneously classify and localize the images to identify the species and the gender of mosquitoes. The authors reported that the concatenated two YOLO V3 models were optimal in identifying the mosquitoes, with a mean average precision and sensitivity of 99% and 92.4%, respectively. In another study, Rustam et al. proposed a system to detect the presence of two critical disease-spreading classes of mosquitoes [9]. The authors introduced a hybrid feature selection method named RIFS, which integrates two feature selection techniques—the Region of Interest (ROI)-based image filtering and the wrapper-based Forward Feature Selection (FFS) technique. The proposed approach outperformed all other models by providing 99.2% accuracy. Further, a lightweight deep learning approach was proposed by Yin et al. for mosquito species and gender classification from wingbeat audio signals [10]. A one-dimensional CNN was applied directly on a low-sample-rate raw audio signal. The model achieved a classification accuracy of over 93%. In [11], the authors proposed a deep learning-based framework for mosquito species identification. The Convolutional Neural Network comprised a multitiered ensemble model. The results demonstrate the model as an accurate, scalable, and practical computer vision solution with 97.04 ± 0.87% classification accuracy. Motta et al. employed a Convolutional Neural Network to accomplish the automated morphological classification of mosquitoes [12]. In their research, the authors compared LeNet, GoogleNet, and AlexNet’s performance and concluded that GoogleNet outperformed all other models, with a detection accuracy of 76.2%. Lastly, Li-Pang, in [13], proposed an automatic framework for the classification of mosquitoes using edge computing and deep learning. The proposed system was implemented with the help of IoT-based devices. The highest detection accuracy that the authors reported was 90.5% on test data.

Robot-assisted surveillance has become an attractive solution for performing various automated tasks. It has widely been used for bringing a certain degree of quality and precision that human labor would be unable to maintain consistently for long periods. In the literature, various robot-assisted applications, such as crawl space inspection, tunnel inspection, drain inspection, and power transmission line fault detection, have been reported. In [14], the authors designed an insect monitoring robot to detect and identify Pyralidae insects. The contours of Asian Pyralidae insect characteristics are selected using the Hu moment feature. The authors reported a recognition rate of 94.3%. In [15], Kim et al. proposed a deep learning-based automatic mosquito sensing and control system for urban mosquito habitats. The Fully Convolutional Network (FCN) and neural network-based regression demonstrated a classification accuracy of 84%.

In [16], the authors proposed Unmanned Aerial Vehicles (UAVs) for identifying malaria vector larval habitats (Nyssorhynchus darlingi) and breeding sites with high-resolution imagery. The results demonstrated that high-resolution multispectral imagery where Nyssorhynchus darlingi is most likely to breed can be obtained with an overall accuracy of 86.73–96.98%. Another study by Dias et al. proposed the autonomous detection of mosquito breeding habitats using a UAV [17]. Here, the authors used a random forest classifier algorithm and reported detection accuracy of 99% on the test dataset.

This work presents an AI-enabled mosquito surveillance and population mapping framework using our in-house-developed robot, Dragonfly. Our research studies entomological characterizations of mosquitoes and obtains the required information to detect, classify, and map various breeds of mosquitoes in various environments. The main objective of the Dragonfly robot developed is to identify the mosquito hotspots and trap and kill mosquitoes by attracting the insects towards it. Moreover, from a pest control management perspective, it is crucial to identify mosquito distribution in a given region to take countermeasures to restrict the infestation effectively. Public health experts can also study the mosquito population and deploy necessary mosquito management programs. These programs help to effectively control the mosquito population and protect humans from life-threatening mosquito-borne diseases. Currently, a real-time deployable robot system for mosquito surveillance infestation control is lacking, which makes our research valuable to the community.

This paper is organized as follows. Section 1 presents an introduction and literature review. Section 2 provides the methodology and an overview of the proposed system. The experimental setup, findings, and discussion are covered in Section 3. Finally, Section 4 concludes this research work.

## 2. Overview of the Proposed System

Figure 1 shows the overview of our deep learning-based robot-assisted mosquito surveillance and population mapping framework. The framework uses our in-house-developed differential drive robot named ‘Dragonfly’, constructed with a UV-powered mosquito trap and a deep learning (DL) computing source to detect and classify the trapped mosquitoes. The details of each module and its description are as follows.

### 2.1. Dragonfly Robot Architecture

This section briefly explains the architecture of the Dragonfly robot. Figure 2 shows a detailed diagram of the robot and Figure 3 shows the system architecture of the Dragonfly robot. The Dragonfly robot is a differential-drive mobile robot base with three points of contact to the ground. Two BLDC motors with integrated velocity control and a passive caster wheel provide mobility and non-holonomic locomotion capability for the robot. Three ultrasonic sensors, downward-pointing infrared cliff sensors, and a hard bumper implemented on the robot offer an additional layer of safety. The robot is equipped with a 2D laser scanner, depth camera, and Inertial Measurement Unit (IMU) sensors that form the primary sensors for perception and navigation. The laser scanner used in the robot is SICK Tim 581 outdoor LiDAR, which provides range information and a 2D obstacle profile for 200 degrees surrounding the robot. An Intel real sense camera (D415i) provides the necessary 3D depth profile of the obstacles and extends the robot’s perception capability beyond the 2D information provided by the LiDAR. The required information for dead reckoning is obtained by fusing wheel odometry computed from the wheel encoders, visual odometry obtained from the real sense camera, and a 9-axis Vectornav IMU. The above-mentioned multi-sensor fusion is achieved by running an extended Kalman filter-based state estimation. The robot performs autonomous navigation on a pre-built map. The global localization is achieved using Adaptive Monte Carlo Localization (AMCL). The robot exploits the A-Star path planning algorithm to perform point-to-point navigation, and a Dynamic Window Approach (DWA)-based local controller for collision avoidance and trajectory tracking.

#### Mosquito Trap Unit

Figure 4 shows the mosquito control unit. The key payload on the Dragonfly robot is a mosquito trap that helps to capture the mosquitoes by attracting them towards an isolated chamber. This unit uses multistage mosquito attractants. These attractants include visual cues, olfactory cues, motion, and a fan. An Ultra Violet (UV) Light-Emitting Diode (LED) at 368 nm is used as a visual cue to lure mosquitoes. Further, chemicals such as octanol and lactic acid are used for a better luring effect. The motion of the robot mimics living things to draw mosquitoes’ attention. In addition, the mosquito robot’s trap is equipped with a fan with a speed of 4.1 m/s that produces a natural airflow. The natural airflow pulls in mosquitoes and confines them inside a special chamber. A yellow observation pad was designed to induce mosquito landing and block the background for easier image processing. The mosquitoes enter the chamber via the downward airflow generated by the fan and stick to the glue card. A pluggable 250× digital USB microscope camera pointing to the glue card is used for mosquito image acquisition. Further, these recorded stream images are used for the mosquito surveillance framework. Here, the Ultra Violet (UV) Light-Emitting Diode (LED) and fan are operated at 24 V. A Teensy microcontroller is used to control the LED and fan. Energy consumption for the entire operation is 3 A.

### 2.2. Mosquito Surveillance Algorithm

YOLO V4 is used for mosquito detection and classification tasks in the surveillance framework. It uses the CSPDarknet53 [18] as a backbone, the neck used is Spatial Pyramid Pooling (SPP) [19] and the Path Aggregation Network (PANet) [20], and the head used is YOLO V3 [21], as seen in Figure 5. Bochkovskiy et al. [22] classified bag of freebies as a method that only changes the training strategy or only increases the training cost; this typically includes data augmentation, regularization, data imbalance, normalization of network activations, the degree of associations between categories, and objective functions of bounding box regressions. Bochkovskiy et al. also a classified bag of specials as a plugin module and post-processing method that only increases the inference cost by a small amount but can significantly improve accuracy; this includes enlarging the receptive field, an attention module, feature integration, and post-processing. Within the bag of freebies and bag of specials that were evaluated, Bochkovskiy et al. introduced 4 modifications: SPP [19], a Spatial Attention Module (SAM) [23], PAN [20], and cross-iteration batch normalization (CBN) [24].

#### 2.2.1. Backbone Architecture

The backbone used is CSPDarknet53; it utilizes cross-stage partial networks, which is an optimization of DenseNet [25]. Partitioning a feature map of the base layers into two parts and then merging them again through a proposed cross-stage hierarchy allows the gradient flow to propagate through different network paths through splitting. As a result, the propagated gradient information can have a more significant correlation difference by switching the concatenation and transition steps, further reducing computational bottlenecks and improving its performance compared to DenseNet. Table 1 shows the CSPDarknet53 backbone with its layer details and input dimensions.

#### 2.2.2. Neck Architecture

For the neck of the mosquito surveillance framework, SPP [19] and PAN [20] are used. Previously, the fully connected layer only allowed for a single input size, resulting in a bottleneck. By utilizing a max pool layer, SPP allows multiple-scale input image training. This results in a larger receptive field size that improves performance at a minimal cost. PAN allows connection between low-level and high-level layers through a shortcut connection via element-wise additions. Instead of using element-wise additions, concatenation is used.

#### 2.2.3. Head Architecture

For the head architecture, YOLO V3 [21] was used. YOLO V3 allows the prediction of boxes at 3 different scales 13×13 for large objects, 26×26 for medium objects, and 52×52 for small objects. For each scale, three different anchors are used. The anchor boxes are determined through the use of k-means clustering. Through this improvement, the total number of predicted boxes for YOLO V3 is 10,647, compared to YOLO V2 with 845 boxes, for an input image of 416×416.

## 3. Experimental Setup and Results

This section describes the experimental results of the mosquito surveillance and population mapping framework. The experiments were carried out in four phases: dataset preparation and training, evaluating the trained mosquito surveillance algorithm model on an offline test, a real-time field trial for mosquito population mapping, and comparing the trained mosquito framework with other models.

### 3.1. Dataset Preparation and Training

The mosquito surveillance framework’s training dataset consists of 500 images of Aedes aegypti mosquitoes [26], 500 images of Aedes albopictus mosquitoes [26], and 500 images of Culex mosquitoes [27]. The dataset consists of a combination of real-time and online collected datasets. Here, the real-time mosquito images were collected using the Dragonfly robot with a mosquito glue trap in gardening regions, a marine dumping yard, and water body areas for real-time mosquito data. Both online collected and real-time collected trap images were resized to 416×416 pixel resolution.

Generally, the mosquito can be trapped or glued on a glue trap in any orientation in the real-time scenario. Thus, data augmentation is applied to the training dataset to overcome the orientation issue. The data augmentation also helps to control the over-fitting and class imbalance issues in the model training stage. Therefore, a total of 15,000 images were used for training. Data augmentation processes such as scaling, rotation, translation, horizontal flip, color enhancement, blurring, brightness, shearing, and cutout were applied to collected images. Figure 6 shows an example of the data augmentation of one image. Table 2 elaborates the settings of the various types of augmentation applied.

#### 3.1.1. Training Hardware and Software Details

The mosquito surveillance algorithm YOLO V4 was built using the Darknet library, and pre-trained CSPDarknet53 was used as a feature extractor [22]. The CSPDarknet53 model was trained on the MSCOCO dataset consisting of 80 classes. The Stochastic Gradient Descent (SGD) optimizer was used to train the YOLO V4 model. The hyper-parameters used were 0.949 for momentum, an initial learning rate of 0.001, and a batch size of 64 along subdivisions of 16. The model was trained for total epochs of 3700 before early stopping and validation of the model in real-time inference.

The model was trained and tested on the Lenovo ThinkStation P510. It consists of an Intel Xeon E5-1630V4 CPU running at 3.7 GHz, 64 GB Random Access Memory (RAM), and an Nvidia Quadro P4000 GPU (1792 Nvidia CUDA Cores and 8 GB GDDR5 memory size running at 192.3 GBps bandwidth).

The K-fold (here K = 10) cross-validation technique was used for validating the dataset and model training accuracy. In this evaluation, the dataset was divided into K subsets; K−1 subsets were used for training, and the remaining subset was used to evaluate the performance. This process was run K times to obtain the detection model’s mean accuracy and other quality metrics. K-fold cross-validation was done to verify that the images reported were accurate and not biased towards a specific dataset split. The images shown were attained from the model with good precision. In this analysis, the model scored 91.5% mean accuracy for K = 10. This indicates that the model was not biased towards a specific dataset split.

### 3.2. Offline Test

The offline test was carried out with augmented and non-augmented images collected using online sources and glue-trapped mosquito images collected via the Dragonfly robot. The model’s performance was evaluated using 50 images composed of three mosquito classes. These images were not used to train the mosquito surveillance framework. Figure 7 shows the mosquito surveillance framework’s experimental results in the offline test, and Table 3 indicates the confusion matrix-based performance analysis results of the offline test experiment.

Here, the algorithm detects mosquitoes with an average confidence level of 88%. Aedes aegypti, Aedes albopictus, and Culex mosquitoes were classified with an accuracy of 78.33%, 77.73%, and 77.81%, respectively, before augmentation. However, Aedes aegypti, Aedes albopictus, and Culex mosquitoes were classified with an accuracy of 93.61%, 90.70%, and 95.29%, respectively, after augmentation. Therefore, it can be concluded that the mosquito surveillance framework demonstrates higher classification accuracy after applying data augmentation. The framework was able to detect and classify most of the mosquitoes. However, the missed detection, false classification, and detection with lower confidence levels were due to partially occluded mosquitoes.

### 3.3. Real-Time Mosquito Surveillance and Mosquito Population Mapping Test

This section evaluates the mosquito surveillance framework’s performance in the real-time field trial. As per the literature survey and NEA Singapore guidelines, mosquitoes are more active at dusk, evening, nighttime, after rainfall, and in environments such as open perimeter drains, covered car parks, roadside drains, and garden landscapes [1,28,29]. Hence, the experiments were carried out during the night (6 p.m. to 10 p.m.) and early morning (4 a.m. to 8 a.m.) in the potential breeding and cluttered environment of the SUTD campus and Brightson ship maintenance facility. In this experiment, the mosquito glue trap was fixed inside the trap unit and performed a mosquito trapping and surveillance function. Figure 8 shows the robot ’Dragonfly’ performing experiments in a different environment. The robot navigated to pre-defined waypoints in the region of operation autonomously on multiple cycles. The robot paused for 10 min at every waypoint, keeping its trap operational to gather more mosquitoes in the respective location. Once the robot completed its navigation to the final waypoint, it moved to the first waypoint and continued its inspection cycle. Figure 9 shows a sample of real-time collected mosquito glue trap images from test environments.

In this real-time analysis, the mosquito glue trap images were captured by a trap camera, and images were transferred to an onboard high-powered GPU-enabled Industrial PC (IPC) for mosquito surveillance and population mapping tasks.

Figure 10 shows the detection results of real-time field trial images, and Table 4 shows the statistical measure results of the mosquito surveillance framework.

The experimental results indicate that the surveillance algorithm detected Aedes aegypti, Aedes albopictus, and Culex mosquito classes on the Dragonfly robot’s captured images with an 82% confidence level. Its bounding region is also accurate with respect to ground truth. The statistical measure indicates that the framework has detected the class of mosquito with a detection accuracy of 87.67% for Aedes aegypti, 86.68% for Aedes albopictus, and 89.62% for Culex mosquitoes. Further, the model’s miss rate is 5.61% for online tests. The missed detection is attributed to mosquito occlusion and blurring due to robot navigation jerking when moving on uneven surfaces.

Figure 11 shows the mosquito population mapping results of the field trial for ten days. The population map was generated by fusing the trapped mosquito classes on a robot navigation map using different color codes. Here, Aedes Aegypti is marked as green, Aedes Albopictus is marked as blue, and Culex is marked in purple. Table 5 shows the details of the number of mosquitoes trapped on a field trial, calculated through the mosquito population mapping function.

From Table 5, it is reported that Aedes Aegypti and Aedes Albopictus are more active during the morning, whereas Culex is primarily active at night. This variation in the number of trapped mosquitoes is because Aegypti and Aedes Albopictus are more attracted to heated objects and covered in body odor. However, in Singapore’s well-lit urban environment, the Aedes mosquito may also be active at night, as it could adapt to artificial lighting [1].

### 3.4. Comparison with Other Existing Model

To evaluate the YOLO V4 model performance, the comparison analysis was performed with three different feature extractors on the YOLO V3 head and SSD MobileNetv2. The training dataset and hardware used were as per Section 3.1 and Section 3.1.1. The hyper-parameters used for MobileNetv2 were 0.9 for momentum, an initial learning rate of 0.08, and a batch size of 128, and images were resized to 640×640. Meanwhile, the hyper-parameters used for ResNet101 were 0.9 for momentum, an initial learning rate of 0.04, and a batch size of 64, and the images were resized to 640×640.

For comparison, a combination of the real-time collected dataset as well as the online collected dataset was used. Fifty images of each class were obtained in real time and online, respectively, resulting in a total of 300 images. Likewise, with the online testing, images from the real-time collection were pre-processed by cropping the images into grids of 416×416 before inference. Table 6 shows the comparison between our proposed model and other object detection models.

The proposed model outperformed the other models in precision, recall, F1, and accuracy. The outliers, being YOLO V3 and ResNet101, were able to outperform in the Aedes aegypti class in terms of precision and F1 score. In terms of FPS, the proposed model managed a decent 57 FPS, ranking third. Specifically for the application on the Dragonfly robot, the trade-off between having higher accuracy is preferred while maintaining a decent FPS.

### 3.5. Comparison with Other Existing Works

This section elaborates the comparative analysis of the proposed algorithm with other existing mosquito detection and classification studies reported in the literature. Table 7 states the accuracy of various inspection models and algorithms based on some similar classes.

The literature has reported various studies focusing on mosquito detection and classification. However, the implementations in these case studies cannot be directly compared to our work. The case studies have employed different training datasets, CNN algorithms, training parameters, and performed offline inspection. Further, the accuracy of our proposed framework is comparatively low, and the proposed framework has a key feature of performing real-time mosquito surveillance and population mapping.

## 4. Conclusions

AI-enabled mosquito surveillance and population mapping was proposed using a Convolutional Neural Network-based framework and in-house-developed robot, ‘Dragonfly’. The Dragonfly robot was designed to capture mosquitoes using multistage mosquito attractants with a differential-drive mechanism. Here, the YOLO V4 DNN algorithm model was trained with three mosquito classes to detect and classify mosquito breeds from a mosquito glue trap. The efficiency of the proposed framework was examined in two phases: an offline test and a real-time mosquito surveillance and mosquito population mapping test. Standard performance metrics including accuracy, precision, recall, and F1 measures were used to assess the drain inspection algorithm. In the comparison analysis, the experiments indicated that the YOLO V4 model outperformed others and was able to detect and classify with an average accuracy of 87.99% in a real-time field trial and process 57 frames per second. Finally, for effective mosquito surveillance, mosquito population mapping was generated by the detection results on the robot’s 2D map. In our future work, we plan on adding more images of different mosquito species in the training dataset to improve the detection and classification accuracy of the surveillance algorithm.

## Figures and Tables

**Figure 1 sensors-22-04921-f001:**
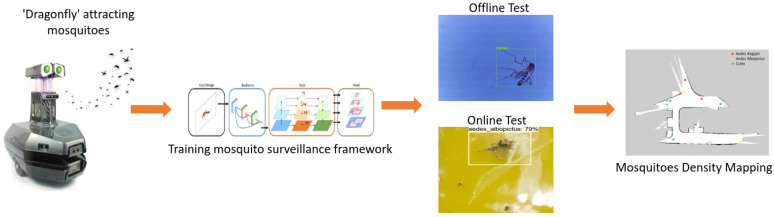
Overview diagram of proposed framework.

**Figure 2 sensors-22-04921-f002:**
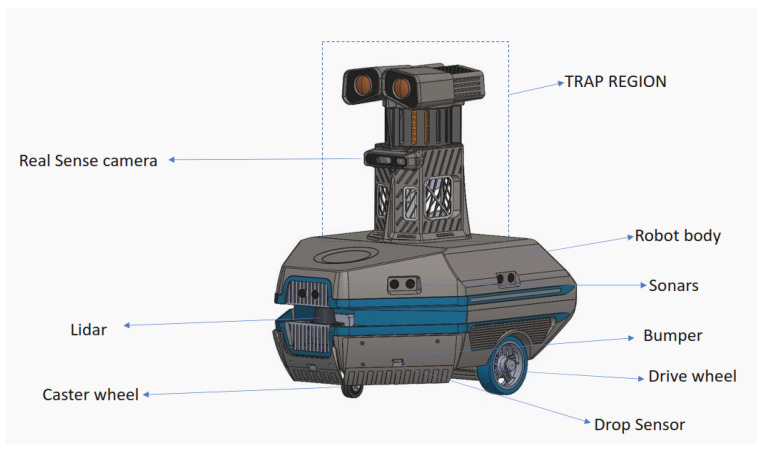
Dragonfly robot.

**Figure 3 sensors-22-04921-f003:**
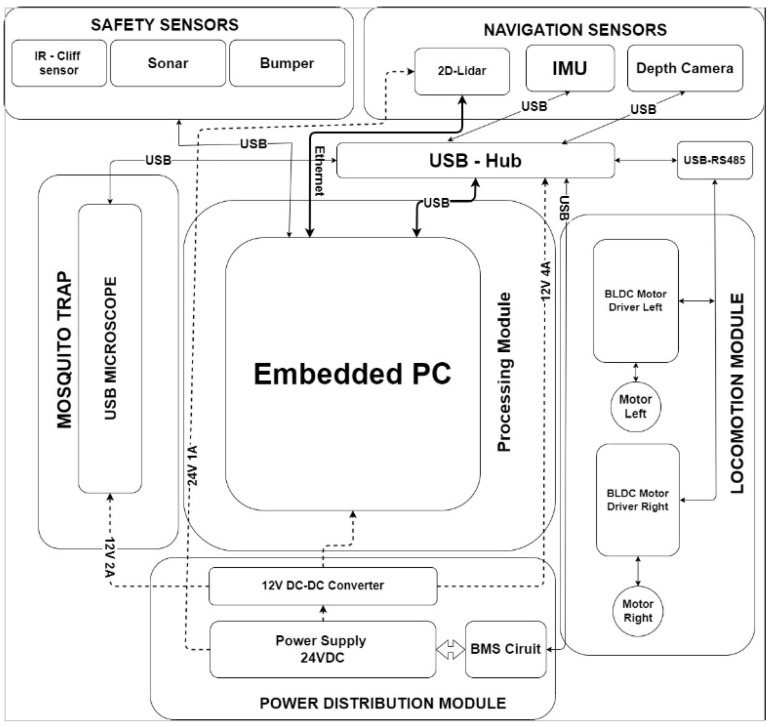
System architecture.

**Figure 4 sensors-22-04921-f004:**
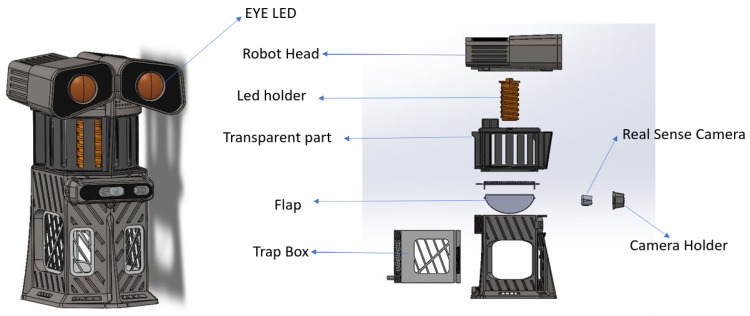
Mosquito trap unit.

**Figure 5 sensors-22-04921-f005:**
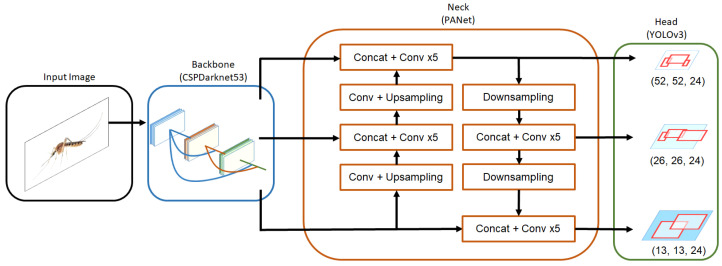
YOLO V4 block diagram.

**Figure 6 sensors-22-04921-f006:**
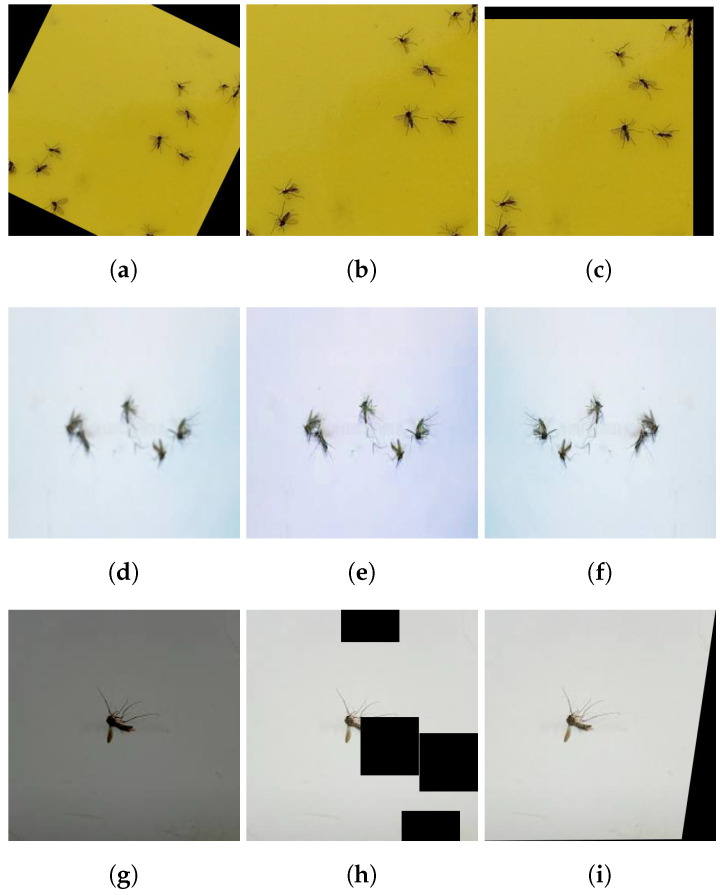
Sample of augmented image. (**a**) Rotation. (**b**) Scale. (**c**) Translation. (**d**) Blur. (**e**) Enhance color. (**f**) Flip. (**g**) Brightness. (**h**) Cutout. (**i**) Shear.

**Figure 7 sensors-22-04921-f007:**
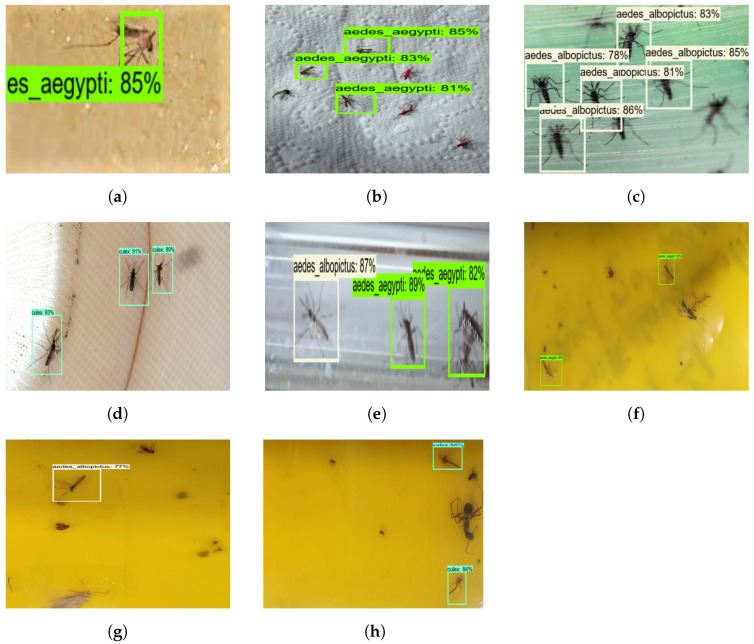
Mosquito surveillance framework’s offline test results. (**a**) Aedes Aegypti. (**b**) Aedes Aegypti. (**c**) Aedes Albopictus. (**d**) Culex. (**e**) Aedes Albopictus and Aedes Aegypti. (**f**) Aedes Aegypti. (**g**) Aedes Albopictus. (**h**) Culex.

**Figure 8 sensors-22-04921-f008:**
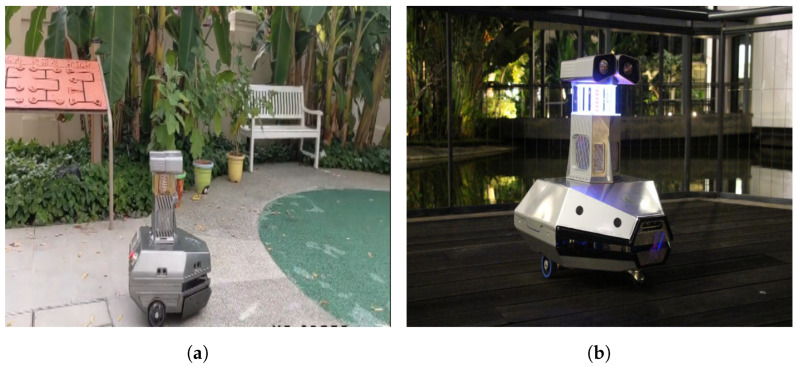
Testing environment. (**a**) ‘Dragonfly’ robot in garden (morning). (**b**) ‘Dragonfly’ robot in SUTD campus (evening).

**Figure 9 sensors-22-04921-f009:**
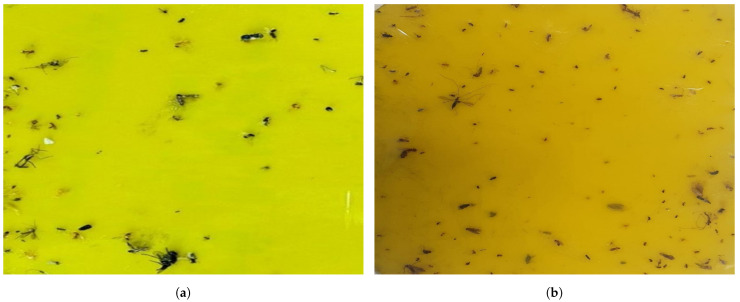
Real-time mosquito glue trap (**a**) Glue Trap 1 (Morning) (**b**) Glue Trap 2 (Evening).

**Figure 10 sensors-22-04921-f010:**
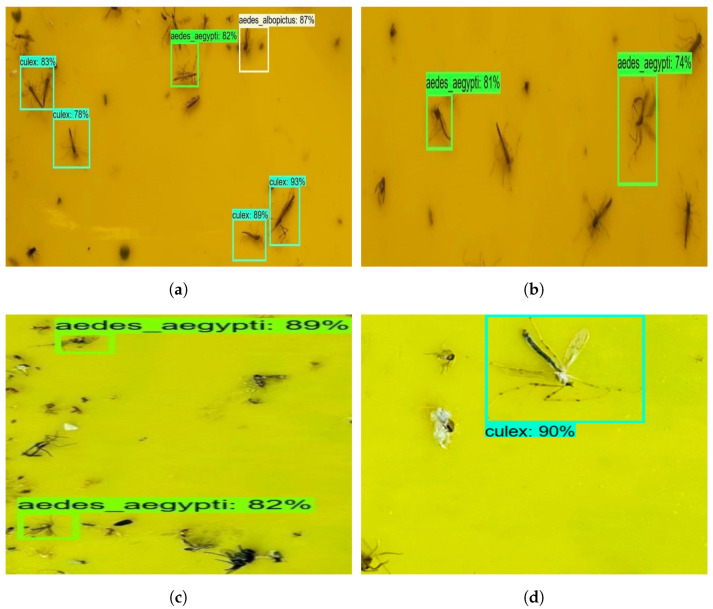
Mosquito surveillance framework’s online test results. (**a**) Culex, Aedes Aegypti, Aedes Albopictus. (**b**) Aedes Aegypti. (**c**) Aedes Aegypti. (**d**) Culex.

**Figure 11 sensors-22-04921-f011:**
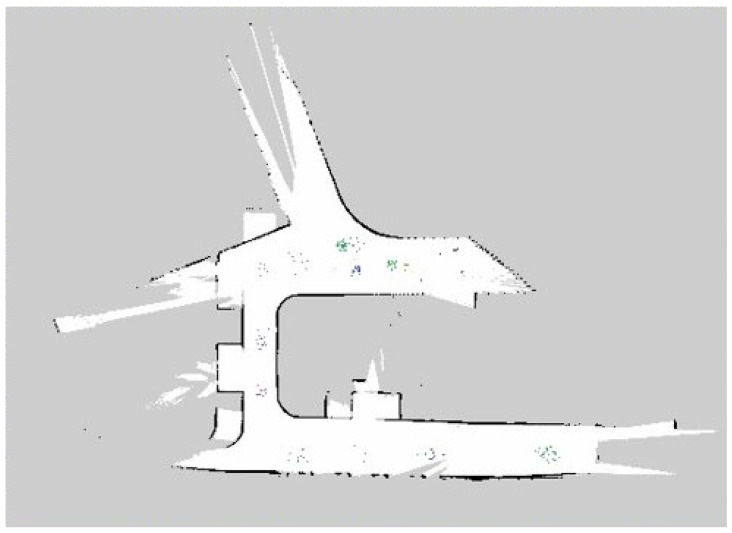
Mosquito population mapping.

**Table 1 sensors-22-04921-t001:** CSPDarknet53 backbone.

Layer Details	Input Dimensions	
Conv	416×416×3	
Conv	32×3×3	
Conv	64×3×3/2	
Conv	32×1×1	×1
Conv	64×3×3	
Residual	-	
Conv	128×3×3/2	
Conv	64×1×1	×2
Conv	128×3×3	
Residual	-	
Conv	256×3×3/2	
Conv	128×1×1	×8
Conv	256×3×3	
Residual	-	
Conv	512×3×3/2	
Conv	256×1×1	×8
Conv	512×3×3	
Residual	-	
Conv	1024×3×3/2	
Conv	512×1×1	×4
Conv	1024×3×3	
Residual	-	
Avgpool	-	
Softmax	1×1×1000	

**Table 2 sensors-22-04921-t002:** Augmentation types and settings.

Augmentation Type	Augmentation Setting
Scaling	0.5× to 1.5×
Rotation	from −45 degree to +45 degree
Translation	*x*-axis (−0.3× to 0.3×) *y*-axis (−0.3× to 0.3×)
Horizontal Flip	.flip the image horizontally
Color Enhancing	contrast (from 0.5× to 1.5×)
Blurring	Gaussian Blur (from sigma 1.0× to 3.0×)
Brightness	from 0.5× to 1.5×
Shear	*x*–axis (−30 to 30) *y*–axis (−30 to 30)
Cutout	1 to 3 squares up to 35% of pixel size

**Table 3 sensors-22-04921-t003:** Statistical measure results of mosquito surveillance framework (offline).

Class	Before Augmentation				After Augmentation			
	Precision	Recall	$F_{1}$	Accuracy	Precision	Recall	$F_{1}$	Accuracy
Aedes Aegypti	73.08	73.44	75.57	76.67	92.25	94.24	93.24	92.67
Aedes Albopictus	73.44	81.03	77.05	77.33	89.51	94.81	92.09	90.00
Culex	75.57	83.90	79.52	78.67	93.66	94.33	94.00	94.00

**Table 4 sensors-22-04921-t004:** Statistical measure results of mosquito surveillance framework (online).

Class	Precision	Recall	*F* _1_	Accuracy	Average Accuracy
Aedes Aegypti	86.30	88.59	87.43	87.67	87.99
Aedes Albopictus	84.90	86.54	85.71	86.68	
Culex	88.52	88.85	88.68	89.62	

**Table 5 sensors-22-04921-t005:** Statistical results of mosquito glue trap.

Class	Early Morning	Night
Aedes Aegypti	54	37
Aedes Albopictus	61	47
Culex	39	64

**Table 6 sensors-22-04921-t006:** Comparison with other models.

Model	Class	Precision	Recall	*F* _1_	Accuracy	Average Accuracy	Inference Speed (FPS)
YOLOv3 + CSPDarknet53 (ours)	Aedes Aegypti	88.29	96.41	92.17	93.61	93.20	57
	Aedes Albopictus	92.83	97.53	95.12	90.70		
	Culex	96.89	98.29	97.59	95.29		
YOLOv3 + ResNet101	Aedes Aegypti	92.28	91.13	93.45	91.67	91.00	24
	Aedes Albopictus	88.77	94.40	91.50	89.33		
	Culex	91.52	93.84	92.67	92.00		
YOLOv3 + MobileNetv2	Aedes Aegypti	81.48	88.00	84.62	83.33	84.22	112
	Aedes Albopictus	82.59	88.14	85.28	84.33		
	Culex	83.58	89.80	86.58	85.00		
SSD + MobileNetv2	Aedes Aegypti	82.90	87.79	85.28	84.67	83.78	98
	Aedes Albopictus	81.18	88.35	84.62	83.00		
	Culex	81.78	87.65	84.61	83.67		

**Table 7 sensors-22-04921-t007:** Comparison analysis with existing object detection frameworks.

Case Studies	Inspection Type	Algorithm	Classes	Accuracy
Rustam et al. [9]	Offline	ETC	2	99.2
Kittichai et al. [8]	Offline	Two YOLO V3	5	99
Yin et al. [10]	Offline	1D-CNN	5	93
Goodwin et al. [11]	Offline	CNN	67	97.04
Proposed framework	Real-time with Dragonfly	YOLO V3 + CSPDarknet53	3	87.99

## Data Availability

The data presented in this study are available on request from the corresponding author.

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
