# Peer review of "AI-Enabled Mosquito Surveillance and Population Mapping Using Dragonfly Robot"

_sensors, 2022, doi:10.3390/s22134921_

Round 1

Reviewer 1 Report

The developed approach in this paper presents artificial intelligence enabled mosquito surveillance and density mapping framework using in-house developed robot,  ’DragonFly’, Yolo V4, DNN algorithm, and 2D environment map generated by the robot. The authors’ findings are original. The manuscript is well written and is comprehensive.

Abbreviations must be spelled out completely on initial appearance in text.

The presented conclusion should be supplemented with the aspects related to the possibility of applying the method in question in further research and in practice. I recommend adding implications to meet Sensors mission and to discuss what is the contribution of this paper to academia, what is the critical issue which should be addressed and how public health experts make better and faster decisions about mosquito management? What are the gaps and limitations of the current research, and what is the contribution of this piece of research to the literature, compared to previous papers?

Reviewer 2 Report

In this study, SLAM was used in conjunction with a mosquito-catching device to monitor mosquito density. This article is quite complete in terms of model comparison, model training methods, model architecture descriptions, and the description and research of the organization's composition. Research topics are also quite interesting. But there are some big problems with the experimental design. Because the experimental design problem, the advice given to the author for this article would be extreme, either "Minor Revise" or "Major Revise". So I select "Minor Revise", but wanted the editor to decide the final decision.

Experiment Design Questions:
1. In the identification of mosquitoes in this paper, offline test and real time identification are inconsistent in experimental design. This reduces the reading value of the overall article.
2. The title is "AI enabled mosquito surveillance and density mapping using Dragonfly robot", which is very suitable for the purpose of this article. However, there are many model comparisons in the content of the article. In terms of identifying mosquitoes, this part is not in line with the concept of "density".
3. In the introduction, there are also many research articles discussing the identification of mosquito species, etc. The structure of the experimental design is in the real time part, and I can't see from the article where it is related to species identification.

Additional Questions:
1. Tables 1 and 2 are missing units.
